# Intermittent Bictegravir/Emtricitabine/Tenofovir Alafenamide Treatment Maintains High Level of Viral Suppression in Virally Suppressed People Living with HIV

**DOI:** 10.3390/jpm13040583

**Published:** 2023-03-27

**Authors:** Baptiste Sellem, Basma Abdi, Minh Lê, Roland Tubiana, Marc-Antoine Valantin, Sophie Seang, Luminita Schneider, Antoine Fayçal, Gilles Peytavin, Cathia Soulié, Anne-Geneviève Marcelin, Christine Katlama, Valérie Pourcher, Romain Palich

**Affiliations:** 1Infectious Diseases Department, Pitié-Salpêtrière Hospital, Sorbonne University, AP-HP, 75013 Paris, France; 2Pierre Louis Epidemiology and Public Health Institute (iPLESP), INSERM 1136, 75013 Paris, France; 3Virology Department, Pitié-Salpêtrière Hospital, Sorbonne University, 75013 Paris, France; 4Pharmacology-Toxicology Department, Bichat Claude Bernard Hospital, AP-HP, INSERM, UMRS 1144, Université de Paris, 75018 Paris, France; 5Pharmacology-Toxicology Department, Bichat Claude Bernard Hospital, AP-HP, IAME, INSERM, UMRS 1137, Université de Paris, 75018 Paris, France

**Keywords:** HIV, antiretroviral therapy, intermittent, people living with HIV

## Abstract

In this observational study, we aimed to evaluate whether bictegravir/emtricitabine/tenofovir alafenamide (B/F/TAF) administered 5 or 4 days a week is able to maintain viral suppression in people living with HIV (PLHIV). We enrolled 85 patients who initiated intermittent B/F/TAF between 28 November 2018 and 30 July 2020: median (IQR) age 52 years (46–59), duration of virological suppression 9 years (3–13), CD4 633/mm^3^ (461–781). Median follow-up was 101 weeks (82–111). The virological success rate (no virological failure [VF]: confirmed plasma viral load [pVL] ≥ 50 copies/mL, or single pVL ≥ 200 copies/mL, or ≥50 copies/mL with ART change) was 100% (95%CI 95.8–100) and the strategy success rate (pVL < 50 copies/mL with no ART regimen change) was 92.9% (95%CI 85.3–97.4) at W48. Two VF occurred at W49 and W70, in 2 patients self-reporting poor compliance. No resistance mutation emerged at time of VF. Eight patients presented strategy discontinuation for adverse events. There was no significant change in the CD4 count, residual viraemia rate, neither body weight during follow-up, but a slight increase in CD4/CD8 ratio (*p* = 0.02). In conclusion, our findings suggest that B/F/TAF administered 5 or 4 days a week could maintain the control of HIV replication in virologically suppressed PLHIV while reducing cumulative exposition of ART.

## 1. Introduction

Although current antiretroviral treatments have improved in terms of efficacy, robustness, genetic barrier to resistance and tolerability, people living with HIV (PLHIV) have remained exposed to drug toxicity for several decades. Since the late 2000s, HIV physicians have been interested in intermittent strategies, i.e., ART administered 5 or 4 days a week, taking advantage of the drug half-life and the viral inertia of HIV to rebound. Several pilot studies, such as the FOTO trial [1] or the ANRS 4D trial [2], have shown that 2 or 3 days off ART may not affect the maintenance of viral suppression. In 2016, the BREATHER randomized trial was proposed to teenagers and young adults treated with efavirenz/lamivudine/tenofovir disoproxil fumarate (EFV/3TC/TDF), administered 5 days a week in the “intermittent arm” [3]. Most recently, the ANRS QUATUOR randomized trial was proposed to adults treated with NNRTI-, INSTI- or boosted PI-based regimens, administered 4 days a week in the “intermittent arm” [4]. These two trials have demonstrated the non-inferiority of intermittent ART in comparison with daily ART. No participant received bictegravir/emtricitabine/tenofovir alafenamide (B/F/TAF), not licenced at time of recruitment in these trials.

In France, intermittent ART has become very popular and common in several settings, since evidence of its effectiveness has been reported in international conferences and the literature. For illustrative purpose, 10.8%, 12.9% and 18.4% of ~4500 PLHIV followed in our centre (Pitié-Salpêtrière hospital, Paris, France) received intermittent ART at the end of 2019, 2020 and 2021, respectively. Here, we aimed to report our experience with intermittent B/F/TAF for maintaining the viral suppression in PLHIV with controlled HIV replication.

## 2. Methods

This observational, non-interventional study included all adults living with HIV who switched to B/F/TAF administered 4 or 5 days a week, from 28 November 2018 to 30 July 2020, in the Infectious Diseases Departement of the Pitié Salpêtrière hospital, in Paris, France. This study was a retrospective analysis, and not a prospective clinical trial. ART prescriptions were made during routine follow-up by HIV physicians, and were not part of any protocol. These prescriptions were discussed with the patients at routine medical visits, as with any change in ART, and respected patient choice. The frequency of treatment was determined by the wording on the electronic prescription (“Biktarvy 1 pill a day 5 days of week” or “Biktarvy 1 pill a day 4 days of week”). Patients were asked to receive the treatment from Monday to Thursday/Friday and not to receive it from Friday/Saturday to Sunday (for the 5 out of 7 day scheme: Monday = ON, Tuesday = ON, Wednesday = ON, Thursday = ON, Friday = ON, Saturday = OFF, Sunday = OFF; for the 4 out of 7 day scheme: Monday = ON, Tuesday = ON, Wednesday = ON, Thursday = ON, Friday = OFF, Saturday = OFF, Sunday = OFF). Compliance with this frequency was checked in the medical records.

Clinical and biological data were routinely collected at baseline and at each medical visit through electronic medical records, collected anonymously in the Nadis database [5], for which all patients provided signed consent (registration number with the “Commission Nationale de l’Informatique et des Libertés, CNIL”: 770134). There were no additional biological samples or questionnaires used for this study. Past HIV-RNA and HIV-DNA resistance genotypes were collected and the cumulative genotype combined all past genotypic tests with an updated interpretation using the latest version of the ANRS algorithm (www.hivfrenchressistance.org) (accessed on 8 February 2023). We also reported antiretroviral plasma concentrations (measured by UPLC-MS/MS [6]) when available, which we linked to days and times of intake, or in the event of VF. Bictegravir plasma concentrations were interpreted according to the protein-adjusted EC_95_ of 162 ng/mL for wild-type HIV-1 [7].

The primary outcome was the proportion of patients with virological success, defined as no virological failure (VF: two consecutive pVL ≥ 50 copies/mL, or isolated single pVL ≥ 200 copies/mL, or single pVL ≥ 50 copies/mL with ART change) at W48. ART change was defined either as a switch to another 3-DR, or the resumption of a daily B/F/TAF regimen for any reason. Secondary outcomes included the proportion of patients with virological success at W96, calculated from data of patients having reached W96, therapeutic success (pVL ≤ 50 copies/mL without ART change) at W48 and W96; the proportion of patients with emergence of genotypic resistance and the plasma antiretroviral drug concentration in the event of VF; the change in CD4 count, CD4/CD8 ratio and weight over the study period, using Wilcoxon tests; the change in the proportion of participants with residual viremia (ultra-sensitive HIV-RNA in the range of 1–20 copies/mL was indicated qualitatively–presence or absence of a detectable signal–using the Cobas AmpliPrep/CobasTaqMan HIV-1 assay, Roche Diagnostics, Switzerland), using a McNemar test.

## 3. Results

A total of 85 patients fulfilling all inclusion criteria switched to an intermittent B/F/TAF strategy, all having reached W48, and 51 having reached W96. Patient characteristics are detailed in Table 1. Median age was 52 years (IQR 46–59), ART duration was 13 years (6–22), and duration of virological suppression was 9 years (3–13). All patients had pVL < 50 copies/mL at study entry. Prior to switching, 79 patients (93%) received an INSTI-based regimen, including 32 (38%) receiving it under daily B/F/TAF started in median 27 weeks (24–38) before. Otherwise, 28 (33%) patients already received an intermittent 3-DR (other than B/F/TAF), including 21 receiving it under elvitegravir/cobicistat/emtricitabine/tenofovir alafenamide.

Median (IQR) follow-up was 101 weeks (82–111). We observed no VFs up to W48, and two VF between W48 and W96, leading to a virological success rate of 100% (95%CI 95.8–100) at W48, and 97.6% (95%CI 91.8–99.7) at W96 (Figure 1).

The two VFs occurred in patients who self-reported suboptimal adherence at time of VF. No genotypic data were available at baseline for both of them. The first VF occurred at W49, with pVL of 332 copies/mL; no antiretroviral plasma concentrations were available at this time, but resumption of daily B/F/TAF led to viral suppression; there was no acquired resistance mutation on HIV genotypic testing. The second VF occurred at W70, with pVL of 758 copies/mL; concomitant plasma concentrations were adequate for bictegravir (C_24h_ = 3524 and C_t_ = 187 ng/mL) and emtricitabine (C_24h_ = 294 and C_t_ < 5 ng/mL), but low for tenofovir (both C_24h_ and C_t_ < 5 ng/mL); resumption of B/F/TAF taken 4 days a week led to viral resupression; there was no acquired resistance mutation on genotypic testing. Eight patients presented a strategy discontinuation between D0 and W96 for adverse events (weight gain n = 3, neuropsychic disorder n = 2, digestive disorder n = 1, renal dysfunction n = 2), leading to a strategy success rate of 92.9% (95%CI 85.3–97.4) at W48, and 89.4% (95%CI 80.8–95.0) at W96.

From D0 to the end of follow-up, there was no significant change in CD4 count: +7 cells/mm^3^ (95%CI −38 to 54, *p* = 0.74), or in the rate of residual viremia: 44% at D0 to 30% at the end of follow-up (*p* = 0.09, N = 66), or in the body weight: +1.2 kg (95%CI −0.8 to 3.0, *p*= 0.26). There was a slight but significant increase in CD4/CD8 ratio: +0.06 (95%CI 0.01 to 0.11, *p* = 0.02).

Plasma antiretroviral concentrations were available for 38 patients, in whom C_24h_ (24 h after the last drug intake) and C_t_ (at the end of the discontinuation window) were measured (Appendix A). Median (IQR) bictegravir, emtricitabine and tenofovir alafenamide plasma C_24h_ were 1598 (1163–2281) ng/mL, 105 (48–305) and 10 (<5–20) ng/mL, respectively; and C_t_ were 114 (68–191), <5 (<5-<5) and <5 (<5-<5), respectively. Bictegravir C_t_ were below the protein-adjusted EC_95_ of 162 ng/mL in 22/38 patients (58%).

## 4. Discussion

The ANRS QUATUOR randomised trial demonstrated the ability of INSTI-based 3-DR to maintain viral suppression in PLHIV with controlled HIV replication [4]. However, the B/F/TAF combination was not studied in this trial. Given its high genetic barrier, good tolerability and low potential for drug interactions, B/F/TAF is now widely used and prescribed as an intermittent regimen in some real-life clinical settings.

In this observational study, we reported a series of 85 patients under B/F/TAF administered 5 or 4 days a week, with high level of virological success (100% at W48 and 97.6% at W96). A majority of those patients were followed for more than two years, and we observed only two VFs, both having occurred in patients with poor adherence. There were no consequences in terms of emerging resistance, and the resumption of B/F/TAF leaded to viral resupression. These data are in line with those obtained in the ANRS QUATUOR trial, which demonstrated that no resistance emerged in case of VF under second generation INSTI-based ART, whether in the “daily arm” or in the “intermittent arm” [4].

The patients presented here had a high CD4 T-cell count (~600/mm^3^) and a sustained viral suppression (~9 years), which could be a favourable profile for reducing treatment pressure. Due to the non-comparative nature of the study, we were not able to compare these immunological and virological parameters with those of PLHIV treated with daily 3-DR. However, it is interesting to note that half of the patients had a CD4 T-cell nadir < 250/mm^3^, and 20% of them had an AIDS-related event in their history.

Nowadays, INSTI-based regimens are positioned as preferred ART for the first treatment in treatment-naive PLHIV in all international guidelines [8]. Several registration trials have proven the high virological potency of B/F/TAF, capable of both suppressing viral replication in the long term, and preventing the emergence of resistance in the event of VF; this combination was well tolerated with less of 6% of adverse events leading to the end of the treatment [9,10,11,12,13]. We assume that offering virally suppressed patients intermittent ART is a reasonable therapeutic option, with the advantage of limiting cumulative treatment exposure, without the need to introduce new drugs, and thus exposing the patient to intolerance.

In general, apart from a clear reduction in the cumulative cost of treatment, we struggle to demonstrate quantitatively the benefit of intermittent regimens for PLHIV. However, in daily practice, patients without comedication report a clear mental health benefit, as the ART-free weekends allow them to briefly forget the burden of the disease. We argue for further studies to demonstrate this using specific scales and qualitative approaches. The reluctance of some HIV doctors is also a barrier to the spread of intermittent ART. Historical studies have shown that non-daily ART is associated with virological failure, virological resistance, immunosuppression, and even mortality [14]. Past antiretroviral combinations were much less robust, and by definition, in the case of poor adherence, ART interruptions are irregular, sometimes very prolonged, and not medically supervised. The “daily versus non-daily” question needs to be reconsidered today in PLWHIV with optimal virological status, taking into consideration the new antiretrovirals.

Antiretroviral drugs with long half-lives are probably the most suitable for use as intermittent regimens. Indeed, the prolonged persistence of the drugs in the body may compensate for the 2–3 day interruption of treatment. The half-life of bictegravir is approximately 17 h, which is the longest half-life compared to other oral INSTIs (dolutegravir: ~12 h, raltegravir and elvitegravir: ~9 h) [15,16]. This prolonged persistence of the drugs in the body allows sufficient plasma concentrations to inhibit viral activity for 2–3 days. Recent data also show the long dissociation of bictegravir from HIV integrase-DNA complexes [17]. This may contribute to more durable intracellular viral activity and resistance to many resistance mutations in comparison to other INSTIs.

Both TAF and INSTIs, including bictegravir, have been incriminated in body weight gain in PLHIV [18,19,20]. One might have expected that the 30–40% reduction in exposure to these two drugs would have limited this effect on body weight, or even resulted in a reduction in weight after the switch. We were unable to demonstrate this, but at least we did not observe any significant weight increase during the study period. Finally, there was no deleterious impact of the intermittent strategy on immunological markers, as previously shown in larger studies [2,4]. We even noted a slight increase in the CD4/CD8 ratio.

The manner in which ART was received by patients was checked at each medical visit by the clinician in charge of the patient, and reported in medical records. The intermittent received was mentioned on the electronic prescription, and in France, the delivery of ART in pharmacies is calculated on the basis of the prescription, so patients receiving intermittent treatment receive boxes of medicines less frequently than patients receiving daily treatment. Differences in plasma drug concentrations from samples collected at the end of rest periods and from samples collected during daily treatment confirmed compliance with the intermittent schedule.

Our study has some limitations, as it is an observational study in a single centre, with a relatively small number of patients included for analysis. Due to the retrospective and observational nature of the study, we had to deal with missing data, which could limit certain analyses. The lack of a comparative arm clearly limits the significance of our findings. However, these original data add to the existing data and support the position of B/F/TAF as a robust antiretroviral combination that could be prescribed intermittently in order to optimize and deliver drug-reduced ART to PLHIV.

## Figures and Tables

**Figure 1 jpm-13-00583-f001:**
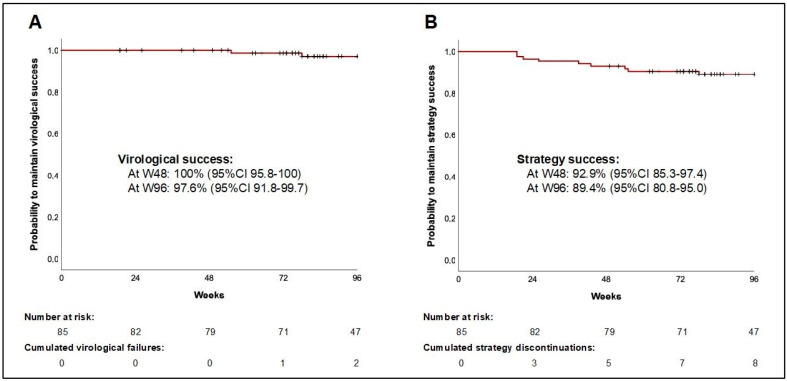
Virological success rate (**A**) and strategy success rate (**B**) under intermittent B/F/TAF over time.

**Table 1 jpm-13-00583-t001:** Patient characteristics at baseline (N = 85).

Age, years, median (IQR)	52 (46–59)
Gender, n/N (%)	
- Male	68/85 (80)
- Female	17/85 (20)
Birth Country, n/N (%)	
- France	58/85 (68)
- Other	27/85 (32)
Transmission group, n/N (%)	
- MSM	53/85 (62)
- Heterosexual	27/85 (32)
- Other	5/85 (6)
Weight, kg, median (IQR)	85 (66–89)
CDC stage C, n/N (%)	17/85 (20)
CD4 nadir, cells/mm^3^ (IQR)	234 (124–338)
Pretherapeutic plasma HIV-RNA, log_10_ cp/mL (IQR)	5 (4.4–5.5)
Time from HIV diagnosis, years, median (IQR)	16 (6–24)
Time from ART initiation, years, median (IQR)	13 (6–22)
HIV subtype, n/N (%)	
- B	32/53 (60)
- Other than B	21/53 (40)
Past resistance to ART, n/N (%) *	
- At least one NRTI	16/55 (29)
- At least one NNRTI	16/51 (31)
- At least one PI	7/53 (13)
- At least one INSTI	1/33 (3)
Genotypic sensitivity score, n/N (%) *	
- 3	28/32 (88)
- 2	1/32 (3)
- 1	3/32 (9)
Duration of viral suppression, years, median (IQR)	9 (3–13)
Residual viraemia, n/N (%)	33/75 (44)
CD4 count/mm^3^ median (IQR)	633 (461–781)
CD4/CD8 ratio, median (IQR)	1.00 (0.65–1.26)
Previous type of antiretroviral strategy, n/N (%)	
- Daily triple therapy	57/85 (67)
- 5 or 4 days a week triple therapy (other than B/F/TAF)	28/85 (33)
Previous type of antiretroviral regimen	
- INSTI-based regimen, n/N (%)	79/85 (93)
- Bictegravir-based regimen, n/N (%)	32/85 (38)
Type of intermittent strategy at inclusion, n/N (%)	
- 5 days a week	39/85 (46)
- 4 days a week	46/85 (54)

NOTES. * Calculated from cumulative historical HIV-RNA and HIV-DNA genotypes with reverse transcriptase, protease and/or integrase available sequences.

## Data Availability

Data can be shared for meta-analysis, for example, upon reasonable request to the corresponding author.

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
