# Peer review of "Intermittent Bictegravir/Emtricitabine/Tenofovir Alafenamide Treatment Maintains High Level of Viral Suppression in Virally Suppressed People Living with HIV"

_jpm, 2023, doi:10.3390/jpm13040583_

Round 1

Reviewer 1 Report

In the manuscript “Intermittent bictegravir/emtricitabine/tenofovir alafenamide treatment maintains high level of viral suppression in virally suppressed people living with HIV” Sellam et al., have investigated the viremia and the CD4/8 T cell status in the people living with HIV, after providing an alternate treatment strategy with intermittently providing the anti-HIV drugs consisting of one integrase inhibitor and two NTRI inhibitors. The study was conducted for significantly long time period of less than two years and the results obtained suggest that the strategy is successful for controlling the viral load and maintaining the T cell count and there is no adverse effects except in patients who with poor compliance. However, the evolution of resistance was also not observed in these patients as well.  The results are presented in a clear manner and the manuscript is presented in standard English language. However, the following points need to be addressed.   

The authors have mentioned the drug half-life in the introduction section as a basis for the intermittent use of these drugs. Whether the integrase inhibitor and NTRIs used in this study have higher half-life in the patient body to compensate for the exemption of drug for 2-3 days in a week, have not been discussed.

The authors have also not compared the virological status and the T cell status in the group of patients treated with a standard regimen of anti-HIV therapy.

It will be better to cite an important article using intermittent therapy.

Hogg RS, Heath K, Bangsberg D, Yip B, Press N, O'Shaughnessy MV, Montaner JS. Intermittent use of triple-combination therapy is predictive of mortality at baseline and after 1 year of follow-up. AIDS. 2002 May 3;16(7):1051-8.

Reviewer 2 Report

Please see attached

Reviewer 3 Report

Sellem et al reported that retrospective studies that aim to evaluate intermittent bictegravir/emtricitabine/tenofovir alafenamide treatment could maintain viral suppression in people living with HIV.

Historically, daily treatment is dispensable for the infected individuals because of low potency of the treatment and prone to virological failure. But the report suggests that INSTI-containing regimens, which exhibit long-acting and a high-genetic barrier to the development of resistance may allow us to treat intermittently.

I don't have major concerns about this manuscript. The following are my comments.

1. Line 85: Please define the detail of the prescription(every 2 days or only weekends...etc). How often did clinicians prescribe "Biktarvy 1 pill a day 4 days of week"? I think the detail will be important. 

2. Line 97: The author mentioned the effective plasma concentration of BIC only. I would suggest adding the information of TAF (TVF) and FTC. One concern is it may become BIC monotherapy if these NRTI/FTC concentration rapidly decay under the effective concentrations.

3. Line 100: Please define the meaning of 'confirmed' and 'single'.

4. For 2 study participants failing treatments, is there any reasons why they  had poor adherence? One concern is that intermitted treatment may lead to missing the drugs. Related to this question, switching to intermittent treatment could improve mental health of patients?
